# Blood Serum and Drainage Microbial and Mitochondrial Metabolites in Patients after Surgery for Pancreatic Cancer

**DOI:** 10.3390/metabo13121198

**Published:** 2023-12-15

**Authors:** Maria Getsina, Nikolay Tsyba, Petr Polyakov, Natalia Beloborodova, Ekaterina Chernevskaya

**Affiliations:** Federal Research and Clinical Center of Intensive Care Medicine and Rehabilitology, 25-2 Petrovka Str., 107031 Moscow, Russia

**Keywords:** GC-MS, phenylcarboxylic acids, aromatic metabolites, pancreatic cancer

## Abstract

Pancreatic cancer (PC) has the highest mortality rate of all major cancers in the world despite improvements in clinical care and an understanding of the biology of pancreatic cancer. A study of 64 patients with verified pancreatic cancer who underwent surgery was included. Sampling was carried out at three points: before surgery and on days 1–3 after surgery and 5–7 days after surgery. Drainage fluid collection was taken from the drains installed intraoperatively one day after surgery. Tyrosine and phenylalanine metabolites and two mitochondrial metabolites, namely succinic and fumaric acids, were identified and quantified by GC-MS in the serum of healthy donors and patients. Differences in the metabolomic profile were found between the patients and healthy people. A statistically significant decrease in the level of *p*-hydroxyphenyllactic acid (*p*-HPhLA), the amount of sum 3 sepsis-associated metabolites (Σ 3AMM), as well as fumaric and succinic acids in patients was observed. It was also noted that *p*-hydroxyphenyllactic acid in the preoperative period may be considered as a predictor of complications and longer postoperative recovery.

## 1. Introduction

Pancreatic cancer (PC) has the highest mortality rate of all major cancers in the world despite improvements in clinical care and an understanding of the biology of pancreatic cancer [1]. Chemotherapy and surgery are the primary treatment options for PC. The high percentage of postoperative complications negatively affect patient survival, causing an increase in the volume of research to develop strategies to reduce them [2,3]. Many of these strategies involve the preoperative risk stratification of patients based on metabolic parameters and various biomarkers [4,5,6,7]. Despite numerous cancer studies using metabolomic approaches and promising results, these studies still highlight the heterogeneity of metabolic features and behaviors used by tumor cells that go beyond the simple rewiring of anabolic and catabolic pathways [8]. Currently, metabolomics is capable of profiling more than 200 small molecule metabolites with important biological functions, with broad coverage of numerous important metabolic pathways, including the tricarboxylic acid cycle (TCA cycle), glycolysis, amino acid metabolism, etc., which are reported to be closely related to the pathogenesis of PC [9,10]. Succinate was recently shown to play a key role in tumorigenesis; it may have both pro- and antitumor efficacy. Therefore, research on succinate may improve our understanding of cancer pathogenesis [11]. In pancreatic cancer, disturbances in amino acid metabolism can lead to changes in the signaling system of molecules responsible for the development of the cancer process, and therefore changes in amino acid profiles can serve as promising diagnostic, prognostic, and therapeutic targets. [12,13]. One of the promising markers for monitoring infectious complications in surgical patients, reflecting microbiota dysfunction, could be aromatic microbial metabolites, but their role in cancer has not been studied [14,15].

The working hypothesis of this study is that metabolomic disorders initially accompany pancreatic cancer and may subsequently cause the development of postoperative complications. The aim of our work is to evaluate the profile of circulating aromatic microbial and mitochondrial metabolites in patients with pancreatic cancer before and after surgery, as well as to identify the possible predictors of postoperative complications.

## 2. Materials and Methods

### 2.1. Study Design

This prospective observational study was performed at the Department of Intensive Care of The Moscow Clinical Scientific Center named after A.S. Loginov, Moscow, Russian Federation. The local Ethics Committee approved the study (N 5/21/3 from 23 December 2021) which was conducted in accordance with the ethical standards of the Declaration of Helsinki and formal consent for participation in this study was obtained from each patient or their legal representative. A study included patients with verified pancreatic cancer who underwent surgery, according to the inclusion criteria.

The inclusion criteria were as follows:Patient with pancreatic cancer;Surgery on the pancreas;Age of patients between 35 and 70.

The exclusion criteria were as follows:Critical condition on admission;Identification of contraindications for surgery;Repeat abdominal surgery

### 2.2. Patients and Samples

Study participants: the inclusion of patients and sample collections are shown in Figure 1.

The study included 64 patients with verified pancreatic cancer. The median age of the patients, stages of cancer, tumor localization, physical status, volume of surgery, and postoperative complications for all patients are shown in Table 1. The physical status of the patients before surgery was calculated according to the American Society of Anesthesiologists (ASA) classification [16], and all postoperative complications were defined according to the Clavien–Dindo classification [17]. One patient did not survive.

Blood serum samples, drainage fluid samples, and clinical data were per the protocol. Samples were taken in patients at three points: point 0—before the surgery (*n* = 64); point 1—on days 1–3 during the early postoperative period (*n* = 64); point 2—on days 5–7 after surgery during the late postoperative period (*n* = 46); and 35 samples of drainage fluid samples in the early postoperative period were also taken. The drainage fluid samples were taken from drains installed intraoperatively in patients one day after surgery (*n* = 35).

The samples of the blood serum from the healthy volunteers (*n* = 48) were collected in Federal State Budgetary Institution N.N. Burdenko Main Military Clinical Hospital (Moscow, Russia). During routine blood sampling in hospitalized patients for biochemical monitoring, the residual serum was taken, frozen, and kept at −30 °C for the analysis of microbial and mitochondrial metabolites by GC-MS.

### 2.3. Reagents

2,3,4,5,6-D5-benzoic acid (surrogate internal standard, ≥99 atom % D, ≥99%), 3-phenylpropionic acid (PhPA, 98%, Sigma-Aldrich, Saint Louis, MO, USA), 3-phenyllactic acid (PhLA, ≥98%), 4-hydroxybenzoic acid (*p*-HBA, ≥99%), 2-(4-hydroxyphenyl)acetic acid (*p*-HPhAA, ≥98%), 3-(4-hydroxyphenyl)propionic acid (*p*-HPhPA, ≥98%), 3-methoxy-4-hydroxyphenylacetic (homovanillic) acid (HVA, ≥97%), 3-(4-hydroxyphenyl)lactic acid (*p*-HPhLA, ≥97%), succinic acid (≥99%), fumaric acid (≥99%), 3,4-dihydroxybenzoic acid (surrogate internal standard), N,O-bis(trimethylsilyl)trifluoroacetamide (99%, contains 1% trimethylchlorosilane), and hexane (≥97.0%) were obtained from Merck (Darmstadt, Germany); ethyl alcohol (95%) were obtained from Kemerovo Pharmaceutical Factory (Kemerovo, Russian Federation); sulfuric acid, acetone, diethyl ether, and sodium chloride were of Laboratory Reagent grade and obtained from Khimreactiv (Kemerovo, Russian Federation).

### 2.4. GC–MS Analysis

All told, 257 blood serum samples of which 174 blood serum samples and 35 drainage fluid from the patient and 48 samples from the healthy volunteers were investigated using the GC-MS method. Samples were defrosted at room temperature prior to use. All GC-MS analyses were performed on a GC-2010 Plus gas chromatograph equipped with an GCMS-QP2020 mass spectrometer (Shimadzu Corporation, Tokyo, Japan) using the capillary column SH-5ms (95% poly(dimethylsiloxane) + 5% phenyl polysilphenylene-siloxane phase, 30 m × 0.25 mm, df = 0.25 µm) obtained from Shimadzu (Shimadzu Corporation, Tokyo, Japan)). The conditions of liquid–liquid extraction of the phenyl carboxylic acids were previously described [18]. Briefly, an aliquot (200 µL) of serum and aliquots (100 µL) of aqueous solution of internal standards (2,3,4,5,6-D5-benzoic, 2,3,3-trideuterio-2-hydroxy-3-(4-hydroxyphenyl)propanoic acid {DL-*p*-hydroxyphenyllactic acid-d3} and 3,4-dihydroxybenzoic acids with a concentration of 4 ng/µL) were diluted with 700 µL of distilled water. Solid sodium chloride (0.3–0.5 g) and concentrated sulfuric acid (15 µL) were added. An extraction with diethyl ether was carried out (2 × 1 mL). The ether extract was evaporated at 40 °C and derivatized with N,O-bis(trimethylsilyl)trifluoroacetamide (20 µL, 80 °C, 15 min). The solution with trimethylsilyl derivatives was cooled at 5 °C for 30 min, diluted with 400 µL of n-hexane, and 2 µL of the final solution was injected into the GC-MS system. Trimethylsilyl derivatives of the phenyl carboxylic acids, succinic and fumaric acids were identified using retention times and characteristic *m*/*z* values which were previously described. The concentrations of the phenyl carboxylic acids, succinic, and fumaric acids were calculated using the equations of linear functions. The linearity of the calibrations was observed in the range of concentration values of 0.5–90 μM. The limit of quantitation for all metabolites was 0.5 µmol/L, and the relative standard deviation <20%.

### 2.5. Biomarkers Analysis

A blood biochemistry test, including levels of amylase and C-reactive protein (CRP), was performed by a chemistry analyzer (AU480, Beckman Coulter, Brea, CA, USA). The biomarker procalcitonin (PCT) was analyzed using electrochemiluminescence (Cobas e411, Roche, Basel, Switzerland).

### 2.6. Statistical Analysis

Data accumulation and initial analysis were conducted using the spreadsheet software “Microsoft Office Excel 2019”. Descriptive statistics for quantitative data were presented in the format Me (Q1; Q3), where Me represents the median value, Q1 is the first quartile (25th percentile), and Q3 is the third quartile (75th percentile). Frequency data were presented as N (%), where N is the absolute number of observations in the group, and % represents the percentage of observations in the group. To assess the conformity of the distribution of the obtained data to the normal distribution, the Shapiro–Wilk test was applied. Due to the non-normal distribution of data for most analyzed parameters, the comparative intergroup analysis for quantitative data was performed using non-parametric statistical tests. The comparison of the two unrelated samples was performed using the Mann–Whitney U test, and for two related variables, the Wilcoxon signed-rank test was used. For the comparative analysis between three groups, the Friedman two-way analysis of variance for related samples was applied, followed by pairwise comparisons using the Dunn–Bonferroni post hoc test. For categorical variables, the comparison of unrelated groups was carried out using the Chi-square test or Fisher’s exact test in cases where the outcome frequency was less than 10%. We used the Kruskal–Wallis test to compare more than two independent groups. The significance of predictors for the studied outcomes was evaluated in ROC analysis, with the presentation of the AUC parameter (area under the ROC curve) and a 95% confidence interval (CI). The optimal cutoff point was selected in ROC analysis according to Youden’s criterion (maximizing the sum of sensitivity and specificity). The critical significance level (*p*) was set at 0.05 (two-tailed). For the visual representation of the distribution of quantitative data, a “box plot” was chosen, which visually displays medians, quartiles, and the range (minimum–maximum). Additionally, pyramid-type charts, bar graphs, and pie charts were used. The statistical analysis of the data obtained during the dissertation research was conducted using SPSS Statistics software (IBM SPSS Statistics for Windows, Version 27.0.1 Armonk, NY, USA: IBM Corp) and MedCalc^®^ Statistical Software version 20.305 (MedCalc Software Ltd., Ostend, Belgium; https://www.medcalc.org). Microsoft Office Excel 2019 was used for creating visualizations and tabular representation of results.

## 3. Results

### 3.1. Metabolites in Healthy Volunteers and Patients before the Surgery

A total of 10 aromatic microbial and mitochondrial metabolites were identified in the serum in patients and compared with healthy donors. The values of some of the determined metabolites, such as: phenylpropionic, phenyllactic, *p*-hydroxyphenylpropionic, homovanillic, *p*-hydroxybenzoic acids were less than the limit of quantitation (<0.5 µmol/L) and are not further discussed in the tables. The results of determining metabolites in healthy volunteers and patients before surgery are shown in Table 2. It can be noted that *p*-hydroxyphenyllactic acid (*p*-HPhLA), Σ 3 AMM, succinic acid, and fumaric acid have higher values in healthy people than in patients.

### 3.2. Changes in the Metabolomic Profile of Patients during the Perioperative Period

Changes in the metabolic profile of patients over time were assessed (Table 3). The values of *p*-HPhLA, Σ 3 AMM, and fumaric acid were statistically significantly different, while the median of *p*-HPhLA and Σ 3 AMM during the preoperative period is higher than after surgery but fumaric acid, on the contrary, was lower.

### 3.3. The Metabolomic Profile in Patients with a Complicated Course during the Postoperative Period

We analyzed the postoperative period taking into account various criteria: the classification according to the Clavien–Dindo complications scale, the length of stay in the ICU, and the presence of infectious complications.

#### 3.3.1. The Clavien–Dindo Classification

Changes in the metabolic profile in patients with postoperative complications shown at Figure 2 and Table A1 Appendix A.

There were no statistically significant differences in the metabolomic profiles of patients before surgery. We identified multiple excesses of CRP levels in patients after surgery (point 1), while the procalcitonin level in these groups of patients remained within the reference values. A statistically significant increase in the level of PhLA and CRP was detected in the group of patients with serious complications compared with minor ones or their absence.

#### 3.3.2. The Length of Stay in the ICU

Patients were divided according to the length of stay in the ICU: short course—less than 3 days; long course—3 or more days. Changes in the metabolic profile are shown in Table 4. During the preoperative period, a statistically significant increase was observed, *p*-HPhLA 0.94 (0.76; 1.22) (*n* = 31) vs. 1.36 (0.98; 1.65) (*n* = 33), (*p* = 0.009) in patients who were in the ICU for more than 3 days. It should also be noted that, in point 1, the number of phenyllactic acid values c > 0.5 in the group of patients with a long course was higher (58%) than in the group of patients with short course (19%). In point 2, the long course patients, the decrease in the values of succinic acid was statistically significantly (*p*-value = 0.039) compared with the short course, 2.91 (2.5; 3.8) (*n* = 31) vs. 2.28 (1.5; 2.9) (*n* = 33), respectively. Regression analysis was performed to account for factors that might influence this difference. In summary, *p*-Hydroxyphenyllactic acid was independently associated with the difference between patients who were in the ICU less than 3 days and those who were in the ICU for 3 days or more (*p*-value = 0.012, adjusted odds ratio = 12.77 (1.75; 93.05)). Such parameters include gender, age, ASA, or tumor stage, which had no statistically significant effect on the differences between the independent groups (Table A2, Appendix A).

ROC analysis showed that the preoperative *p*-HPhLA level was a statistically significant predictor of prolonged stay in the ICU (for 3 days or more) of average quality (*p*-value = 0.005; AUC = 0.689 (0.557; 0.821); Youden’s index: >1.10 (>0.90; >1.5); sensitivity = 69.7%; specificity = 67.7%.

A hospital stay of more than 12 days was considered as a criterion that makes it possible to retrospectively assess the characteristic changes in the metabolic profile. Benzoic acid in the group 5–7 days decreased in a statistically significant manner after surgery, 0.77 (0.65; 1.1) (*n* = 28) vs. 0.6 (<0.5; 0.71) (*n* = 36) (*p*-value = 0.005). No other statistically significant differences in the metabolic profile were found.

#### 3.3.3. Infectious Complications

Patients with PCT values < 0.25 were assigned to the group without inflammation and infection. A statistically significant increase in the concentration of *p*-HPhLA 1.1 (0.9; 1.4) (*n* = 57) vs. 1.8 (1.3; 1.9) (*n* = 8) (*p*-value = 0.003) was revealed in patients with elevated procalcitonin values in the preoperative period.

The comprehensive assessment of infectious complications in the postoperative period, namely the presence of inflammation (body temperature, heart rate, tachypnea, leukocytes), microbiology test, an additional course of antibiotics, additional invasive intervention, increased PCT values, and CRP did not reveal statistically significant differences in the metabolomic profile at all study points.

#### 3.3.4. Chemotherapy and Extent of Surgery

There was no effect of prior chemotherapy on the metabolomic profile of patients with chemotherapy (*n* = 48) compared to the patients without it (*n* = 16).

The influence of the volume of pancreatic resection was analyzed; no differences were identified in the metabolomic profile at points 0 and 1; however, at point 2, there were statistically significant differences in Σ 3 AMM in patients with pancreaticoduodenectomy/gastropancreatoduodenectomy + total duodenopancreatectomy (*n* = 48) vs. distal pancreatectomy (*n* = 16), namely 2.4 (1.8; 3.9) vs. 1.71 (1.5; 1.9) (*p*-value = 0.031), respectively.

### 3.4. Metabolomic Profile in Drainage Fluid

The metabolomic profile of drainage fluid has not been previously studied and reference values are not known. There was a statistically significant difference in the values in the blood serum of patients on days 1–3 after surgery in comparison with the drainage fluid of such metabolites as: benzoic acid 0.58 (0.41; 0.72) vs. 23 (8.8; 47) (*p*-value < 0.001); succinic acid 2.4 (1.7; 3.1) vs. 5.5 (3.8; 8.9) (*p*-value < 0.001); fumaric acid 0.54 (0.40; 0.66) vs. 2.9 (1.6; 4.9) (*p*-value < 0.001) (Table 5).

The benzoic acid level (median) was higher in the drainage sample when the amylase level was greater than 300–36 (11; 53), *n* = 18 versus 21 (9; 37) µmol/L (*p*-value = 0.231), *n* = 17, respectively.

Moreover, statistically significant differences in the drainage of patients undergoing chemotherapy before surgery were revealed compared with the group without previous chemotherapy with regard to the level of *p*-HPhLA 1.9 (1.4; 2.5) vs. 1.2 (0.84; 1.8) (*p*-value = 0.028) and Σ 3 AMM 2.8 (2.1; 4.3) vs. 2.1 (1.4; 2.5) (*p*-value = 0.025), respectively.

## 4. Discussion

The study focuses on changes in the metabolomic profile in the serum samples of patients with pancreatic cancer. Differences in the metabolomic profile were found between patients and healthy people. A statistically significant decrease in the level of *p*-HPhLA and Σ 3 AMM, as well as fumaric and succinic acids, in patients was observed. The subject of many studies is that the reduction in metabolites indicates a change in the microbial load in patients with pancreatic cancer. The composition of the microbiota can be influenced not only by pathogenetic changes [19], but also by treatment, including chemotherapy and antibiotic therapy before or after surgery. Changes in the metabolomic profile are considered in connection with preoperative factors, such as “antibiotic therapy” and “chemotherapy”, which may indirectly reflect the severity of the condition and the possibility of developing infectious complications. The search for predictors can be attributed to two directions: preventing the development of infectious complications during the early postoperative period and assessing the rate of recovery after surgery. However, no significant changes in the metabolic profile associated with infectious complications were found in this cohort of patients.

Currently, the most progressive way to optimize the pre- and postoperative period is the concept of fast-track surgery [20,21]. Changes in the metabolomic profile before surgery, and in the early and late periods after surgery, can be assessed as not significant. Parameters such as “length of stay in the ICU” and “length of hospital stay” can be considered as factors influencing the rate of recovery after surgery. The preoperative *p*-HPhLA level was a statistically significant predictor with a prolonged ICU stay (3 days or more) of average quality (*p*-value = 0.005; AUC = 0.689 (0.557; 0.821). *p*-HPhLA is described as having prognostic significance in cases of infectious complications and sepsis; its values can increase ten-fold. [14]. Meta-analysis showed that the risk of death was 1.5 times higher in patients experiencing severe postoperative complications (Clavien–Dindo ≥ III) than in patients without severe complications [3]. We identified a statistically significant increase in the level of PhLA in the group of patients with severe complications according to the Clavien–Dindo scale with a grade of III–V, which was also accompanied by higher C-reactive protein values. It was previously discovered that PLA can be produced by a wide range of Gram-positive microorganism species, such as *Lactobacillus*, *Enterococcus*, *Weissella*, and *Leuconostoc*, but the production varied greatly among strains and species [22,23,24]. A multiple increase in the level of PhLA in the serum of patients with sepsis was established [25]. This presumably confirms the role of microbiota dysfunction in the development of severe infectious and non-infectious complications.

A frequent and severe complication in pancreatic surgery is postoperative pancreatic fistula [26]. One of the early diagnostic methods was that of testing the drainage fluid amylase concentrations, which are more or equal than 3 times the upper limit of normal serum value [27]. Surgical drains are artificial tubes used to remove blood, pus, or other fluids from the body. A closed drain consists of a tube that drains into a waste collection bag. Drainage is typically used after pancreatic surgery because of the potential for the accumulation of bile, drainage fluid, or blood, which may cause additional invasive procedures [28]. But the optimal cutoff value varies among study centers and the results are still controversial [29,30]. Although the metabolic profile of the drainage fluid in this study was different from that of the serum, we did not find significant differences in the patients above the amylase thresholds other than a high BA concentration. We assumed that this trend was due to the accumulation of benzoic acid during complications as a result of microbial production [31], but there was no microbiological confirmation. Perhaps further research will clarify the diagnostic ability of this metabolite.

It can be assumed that the absence of high values of metabolites in serum is associated with a change in the composition of the taxonomic microbiota. Bacterial species associated with pancreatic cancer such as *Enterobacteriaceae*, *Enterococcus faecalis*, *H. Pylori*, *Porphyromonas gingivalis*, *Fusobacterium* spp., *Bifidobacteria*, *Bacteroides,* and a number of bacterial metabolites such as volatile fatty acids and lipopolysaccharides are associated with carcinogenesis and with the inflammation and immune response. In the treatment of pancreatic cancer, the intestinal microbiota undergoes changes when exposed to chemotherapy and immunotherapy [32]. A change in the postoperative composition of the intestinal microbiota, also described in cases of colorectal cancer, was noted in the work, which can lead to increased virulence, the proliferation of pathogens, and a decrease in the number of beneficial microorganisms, leading to severe complications, including anastomosis failure and infections in the surgical intervention area [33].

Our study is a pilot study because the number of patients included in the study is not large. Unfortunately, we did not make the sample power necessary for our study in advance. The calculated sample power obtained during the calculation in this study was 73%. It should be noted that the sample is rather heterogeneous, as patients differ both in the degree of invasiveness of the operations performed, in the antibiotic therapy carried out in both the preoperative and postoperative periods in concomitant diseases. There were also no differences in the stages of cancer due to a small sample of patients. Given the severity of this pathology, it is necessary to evaluate not only changes in the metabolomic profile which provides indirect data on intestinal dysbiosis, but also to evaluate the changes in more detail, which may be a further direction of research.

## 5. Conclusions

Changes in the metabolomic profile in patients with pancreatic cancer before and after surgery are considered. The concentrations of 10 metabolites were determined by GC-MS in the blood serum of healthy donors and patients, including three sepsis-associated aromatic microbial metabolites such as PhLA, *p*-hydroxyphenylacetic acid, and *p*-HPhLA. Differences in the metabolomic profile were found between patients and healthy people. A statistically significant decrease in the level of *p*-HPhLA, the amount of Σ 3 AMM before surgery, as well as fumaric and succinic acids in patients was observed. *p*-HPhLA during the preoperative period can be considered as a predictor of complications and longer postoperative recovery. The high level of PhLA during the early postoperative period may be considered a predictor of severe complications. It was noted that the extent of resection, tumor localization, and the presence of additional invasive intervention do not affect the metabolic profile.

## Figures and Tables

**Figure 1 metabolites-13-01198-f001:**
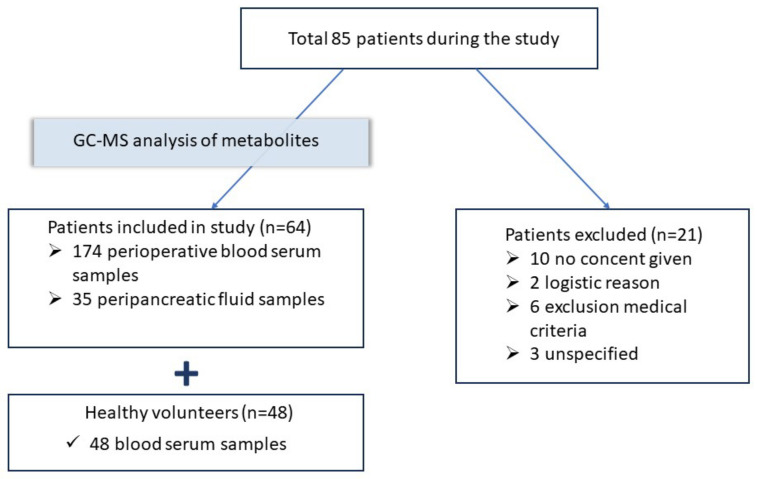
The study design: 64 patients were included in the study.

**Figure 2 metabolites-13-01198-f002:**
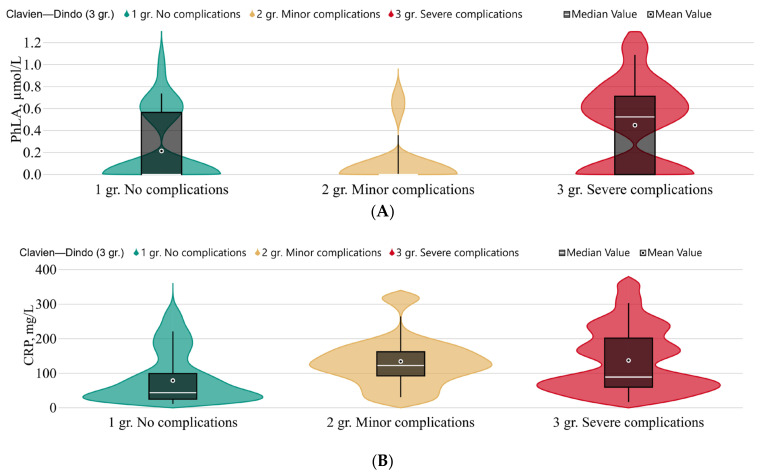
The concentrations of: (**A**) phenyllactic acid, µmol/L; and (**B**) **C**-reactive protein (CRP), mg/L according to Clavien–Dindo classification complications scale in point 1. Green—no complications (*n* = 28); yellow—minor complications (grades I–II) (*n* = 10); red—severe complications (grades III–V) (*n* = 24).

**Table 1 metabolites-13-01198-t001:** Parameters of healthy volunteers and patients included in the study.

Parameter	Healthy Volunteers (*n* = 48)	Patients before the Surgery (*n* = 64)
Sex, male, %	35, 72%	31, 48%
Age, years	40 (34; 45)	60 (54; 70)
Stages of cancer, *n*, (%)	-	0—5, (8%); I—29, (45%), II—24, (58%), III—24, (8%), IV—1 (1%)
Tumor localization, *n*, (%)	-	Head of the pancreas—29 (46%), Body/tail of the pancreas—20 (31%), Major duodenal papilla—13 (20%), Terminal cholidochus—1 (1.5%), Total—1 (1.5%)
Systemic pathology, *n*, (%)	-	No—15 (23.5%), Cardiovascular system—29 (45%), Respiratory—1 (1.5%), Endocrine—3 (5%), Digestive—3 (5%), Polymorbidity—13 (20%)
American Society of Anesthesiologists (ASA), class, (%)	-	II—36, (56%), III—24, (38%), IV—4, (6%)
Volume of surgery, *n*, (%)	-	Gastropancreatoduodal resection—45 (70%), Distal resection of the pancreas—16 (25%), Duodenumpancreatectomy—3 (5%)
Postoperative complications, *n* cases, (%)	-	Acute fluid accumulation—20 (29%); Bleeding—7 (11%); Pancreatitis—23 (36%); Anastamosis failure—6 (9%); Sepsis/MOF—1 (2%); Thrombosis—2 (4%); Peritonitis—2 (4%); Pneumonia—3 (5%)

**Table 2 metabolites-13-01198-t002:** The concentrations of aromatic and mitochondrial metabolites in the serum samples of the patients before the surgery (*n* = 64) and healthy volunteers (*n* = 48), and the results of the Mann–Whitney U test.

Metabolites, µmol/L	Healthy Volunteers (*n* = 48)	Patients before the Surgery (*n* = 64)	*p*-Value ***
Benzoic acid	0.5 (0.5; 0.6)	0.6 (<0.5; 0.6)	-
Phenylpropionic acid	<0.5 (<0.5; <0.5)	<0.5 (<0.5; <0.5)	-
*p*-Hydroxyphenylacetic acid	<0.5 (<0.5; <0.5)	<0.5 (<0.5; 0.6)	-
*p*-Hydroxyphenyllactic acid	1.3 (1.0; 1.6)	1.1 (0.9; 1.4)	**0.002**
Σ 3 AMM *	1.9 (1.5; 2.2)	1.4 (0.9; 2.0)	**<0.001**
Succinic acid	4.8 (4.4; 5.9)	2.9 (2.2; 3.9)	**<0.001**
Fumaric acid	1.3 (1.1; 1.5)	0.6 (0.5; 0.9)	**<0.001**
CRP (mg/L)	<5 **	4.3 (1.8; 14.8)	-

* Σ 3 AMM the sum of three sepsis-associated aromatic microbial metabolites, which consists of phenyllactic (PhLA), *p*-hydroxyphenylacetic, and *p*-hydroxyphenyllactic acids (*p*-HPhLA). ** reference range. *** values are in bold if *p*-value < 0.05.

**Table 3 metabolites-13-01198-t003:** The concentrations of aromatic and mitochondrial metabolites, PCT, and the results of the Friedman test. Samples were taken in patients at three points: point 0—before surgery (*n* = 64); point 1—on days 1–3 during the early postoperative period (*n* = 64); point 2—on days 5–7 after surgery during the late postoperative period (*n* = 46).

Metabolites, µmol/L	Point 0	Point 1	Point 2	*p*-Value *
Benzoic acid	0.59 (<0.5; 0.74)	0.58 (<0.5; 0.72)	0.68 (0.51; 0.78)	0.345
Phenylpropionic acid	<0.5 (<0.5; 0.5)	<0.5 (<0.5; <0.5)	<0.5 (<0.5; <0.5)	-
Phenyllactic acid	<0.5 (<0.5; <0.5)	<0.5 (<0.5; 0,62)	<0.5 (<0.5; 0.60)	-
*p*-Hydroxyphenylacetic acid	<0.5 (<0.5; 0.64)	<0.5 (<0.5; 0.89)	<0.5 (<0.5; 1.8)	-
*p*-Hydroxyphenyllactic acid	1.1 (0.87; 1.5)	1.3 (0.98; 1.9)	1.2 (0.98; 1.7)	**<0.001**
Σ 3 AMM	1.9 (1.5; 2.4)	2.4 (1.7; 3.2)	2.2 (1.7; 3.8)	**0.005**
Succinic acid	2.9 (2.2; 3.9)	2.3 (1.7; 3.1)	2.8 (1.7; 3.4)	0.056
Fumaric acid	0.63 (0.50; 0.94)	0.54 (<0.5; 0.66)	0.49 (<0.5; 0.68)	**<0.001**
PCT	0.054 (0.022; 0.092)	0.14 (0.078; 0.37)	0.10 (0.074; 0.19)	**<0.001**

* values are in bold if *p*-value < 0.05.

**Table 4 metabolites-13-01198-t004:** The concentrations of aromatic and mitochondrial metabolites in the blood serum of patients before surgery: short course—stay in the ICU from 0 to 3 days (*n* = 31); long course—stay in the ICU for 3 days or more (*n* = 33), and the results of the Mann–Whitney U test.

Metabolites before Surgery, µmol/L		Patients Days in the ICU Less than 3 (*n* = 31)	Patients Days in the ICU 3 Days or More (*n* = 33)	*p*-Value *
Benzoic acid		0.55 (<0.5; 0.76)	0.62 (<0.5; 0.68)	0.614
*p*-Hydroxyphenyllactic acid		0.94 (0.76; 1.2)	1.4 (0.98; 1.6)	**0.009**
Σ 3 AMM		1.7 (1.4; 2.4)	2.2 (1.6; 2.4)	0.087
Succinic acid		2.6 (2.2; 4.0)	3.1 (2.3; 3.7)	0.559
Fumaric acid		0.71 (0.53; 0.94)	0.59 (<0.5; 0.91)	0.440
General				
Sex (M)		12 (38.7%)	19 (57.6%)	0.131
Age		59.0 (53.0; 65.0)	65.0 (54.0; 72.0)	
ASA	2	21 (67.7%)	15 (45.5%)	0.059
	3	10 (32.3%)	14 (42.4%)	
	4	0 (0.0%)	4 (12.1%)	
Stages of cancer	0	3 (9.7%)	2 (6.1%)	0.846
	1	13 (41.9%)	16 (48.5%)	
	2	11 (35.5%)	13 (39.4%)	
	3	3 (9.7%)	2 (6.1%)	
	4	1 (3.2%)	0 (0%)	
Tumor localization	Head of the pancreas	13 (41.9%)	17 (51.5%)	**0.001**
	Body/tail of the pancreas	17 (54.8%)	2 (6.1%)	
	Major duodenal papilla	0 (0%)	1 (3%)	
	Terminal cholidochus	1 (3.2%)	12 (36.4%)	
	Total	0 (0%)	1 (3%)	

* values are in bold if *p*-value < 0.05.

**Table 5 metabolites-13-01198-t005:** The concentrations of aromatic and mitochondrial metabolites and amylase in serum (*n* = 64) and drainage fluid in point 1 (*n* = 35) and the results of the Wilcoxon test.

Metabolites, µmol/L	Serum (*n* = 64)	Drainage Fluid (*n* = 35)	*p*-Value *
Benzoic acid	0.58 (<0.5; 0.72)	23(8.8; 47)	**<0.001**
Phenyllactic acid	<0.5 (<0.5; 0.6)	<0.5 (<0.5; 0.5)	-
*p*-Hydroxyphenylacetic acid	<0.5 (<0.5; 0.88)	<0.5 (<0.5; 0.6)	-
*p*-Hydroxyphenyllactic acid	1.3 (0.98; 1.9)	1.8 (1.2; 2.4)	0.072
Σ 3 AMM	2.4 (1.7; 3.2)	2.6 (1.8; 3.9)	0.351
Succinic acid	2.4 (1.7; 3.1)	5.5 (3.8; 8.9)	**<0.001**
Fumaric acid	0.54 (<0.5; 0.66)	2.9 (1.6; 4.9)	**<0.001**
Amylase, Unit/L	68.6 (18.1; 156.5)	1003.7 (52.3; 6337.7)	**<0.001**

* values are in bold if *p*-value < 0.05.

## Data Availability

The data presented in this study are available on request from the corresponding author. Data are not publicly available due to privacy.

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
