# Peer review of "Blood Serum and Drainage Microbial and Mitochondrial Metabolites in Patients after Surgery for Pancreatic Cancer"

_metabolites, 2023, doi:10.3390/metabo13121198_

Round 1

Reviewer 1 Report

Comments and Suggestions for Authors

the research framework is very interesting and the authors fully respond to it. pancreatic cancer is fast developing with very low chances for recovery. the understanding of the metabolite's changes during their development can lead to early diagnostics and improve chances for recovery. the introduction is concise and precise with enough information about similar research. the materials and methods are clearly described with all important information and the reader can easily repeat it. the results are presented well and the researcher can make conclusions very easy. the discussion summarizes all the findings and gives a logical explanation for the obtained results. the number of figures and tables is adequate and improves the quality of the paper. the conclusion is supported by the results.

reading the article I have one question for the authors: " Did you try determination of the fumaric and succinic acids by Ion chromatography with MS or CD detection?

Comments on the Quality of English Language

"the English style and grammar are satisfactory without any notable remarks

Author Response

Thank you for your high assessment of our work. To answer your question: We have not tried to determine succinic and fumaric acids by other methods. We are aimed at identifying different classes of compounds in one analysis; the GC-MS method allows us to solve this problem.

Reviewer 2 Report

Comments and Suggestions for Authors

-Summary: The paper presents a metabolomic analysis of blood serum and drainage fluid samples from patients with pancreatic cancer before and after surgery. The authors identified several metabolites that differed significantly between patients and healthy volunteers, as well as between patients with different postoperative outcomes. The paper is well-written, informative, and relevant to the journal's scope.

-Strengths: The paper has several strengths, such as:

    - The use of GC-MS to measure 10 aromatic microbial and mitochondrial metabolites, which are potentially involved in the pathogenesis and prognosis of pancreatic cancer.

    - The comparison of metabolomic profiles between patients and healthy volunteers, as well as between patients with different postoperative complications, length of stay in the ICU, and prior chemotherapy.

    - The identification of p-hydroxyphenyllactic acid (p-HPhLA) as a predictor of prolonged ICU stay and severe complications, and phenyllactic acid (PhLA) as a marker of severe complications according to the Clavien-Dindo classification.

    - The discussion of the possible role of microbiota dysfunction and succinate in the development of postoperative complications and recovery.

-Weaknesses: The paper also has some weaknesses, such as:

    - The small and heterogeneous sample size of patients, which limits the generalizability and statistical power of the results.

    - The lack of microbiological confirmation of the origin and significance of the metabolites measured in the drainage fluid samples.

    - The lack of adjustment for potential confounding factors, such as age, gender, tumor stage, and location, comorbidities, and antibiotic therapy, in the analysis of metabolomic differences between groups.

    - The lack of validation of the findings in an independent cohort of patients or using other analytical methods.

-Suggestions: The paper could be accepted. But possible improvement are as follows;

    - Increase the sample size and stratify the analysis by relevant clinical and demographic characteristics of the patients.

    - Perform microbiological analysis of the drainage fluid samples and correlate the results with the metabolomic profile and the postoperative outcomes.

    - Adjust the statistical analysis for potential confounding factors and report the effect sizes and confidence intervals of the metabolomic differences between groups.

    - Validate the findings in an independent cohort of patients or using other analytical methods, such as NMR or LC-MS.

Author Response

Dear reviewer, thank you for your valuable comments, which increased the significance of our research. We agree that the paper also has some weaknesses, and have tried to correct them:

    - Increase the sample size and stratify the analysis by relevant clinical and demographic characteristics of the patients.

The study was designed as a single-center pilot study, which has limitations (lines 354–363). We cannot increase the sample size for this article, but we plan to continue our research. We stratify the patients by relevant clinical and demographic characteristics, such as gender, age, stage of cancer, tumor location.  It’s did not affect our results, except for the location of the tumor, which is reflected in table 4, in the text on lines 234-239 and in appendix tables 1 and 2.

    - Perform microbiological analysis of the drainage fluid samples and correlate the results with the metabolomic profile and the postoperative outcomes.

Were obtained 9 positive results of drainage fluid cultures, but in some cases, analysis data is not available. There was no correlation between the metabolomic profile, drain amylase levels, postoperative outcomes and microbiological data, so we did not include this information in the main text of the article. It is possible that increasing the number of results or using PCR analysis of drainage fluid will reveal significant results. 

  - Adjust the statistical analysis for potential confounding factors and report the effect sizes and confidence intervals of the metabolomic differences between groups.

We considered this suggestion and performed multivariate regression analysis to identify an independent predictor of duration of ICU hospitalization beyond 3 days (Appendix Table 2. A.).

    - Validate the findings in an independent cohort of patients or using other analytical methods, such as NMR or LC-MS.

The study was designed as a single-center pilot study, but we will take your wishes into account and possibly expand it an independent cohort of patients. On a cohort of healthy volunteers, another our study used this method in comparison with LC-MS. It has been shown that the results are comparable to the GC-MS and LC-MS methods. The results were published in this article: https://www.mdpi.com/2218-1989/13/11/1128.

Reviewer 3 Report

Comments and Suggestions for Authors

The manuscript "Blood serum and drainage microbial and mitochondrial metabolites in patients after surgery for pancreatic cancer" aims to predict postoperative complications in patients with pancreatic cancer by using microbial and mitochondrial metabolites measured from serum and abdominal drainage as potential biomarkers. The authors compare metabolite measurements before and after surgery and test if significant differences in metabolite concentrations associate with complications after surgery. The authors use of metabolites measured from drainage fluid is a novel approach to predict surgical complications and the formation of pancreatic fistulas. The authors contribute differences in key metabolites to alterations in gut microbiota. This study provides an interesting approach to predict surgical complications and identifies key metabolites in a research field that currently lacks reliable biomarkers. However, there are multiple issues that need addressed to enhance this manuscript. My comments are listed.

1. Understanding the patient variables in the methods (Lines 72-79) takes a lot of time when listed in paragraph form. Consider putting the data contained in lines 72-79 in table format. This would make it faster for a reader to interpret. 

2. While the authors mention the small sample size of patients in this study, metabolite measurements (Table 4) and complication severity data (Figure 2) separated into male and female groups would enhance the manuscript. For instance, overall elevated PhLA levels in patients are associated with severe complications (Figure 2) and increased ICU stay (Table 4), it would be interesting to know the sex distribution of these groups too. 

3. In addition to Comment 2, it would also be interesting to know if patient physical status before surgery (ASA categorization) is associated with complications after surgery and/or ICU stay. This would enhance the study since complicating factors such as obesity could have an effect on these data.

4. Do the authors have any additional patient data including preoperative body composition measurements (eg., CT scans) or 1, 3, or 6-month weight loss prior to surgery? Previous studies have correlated >5% body weight loss up to 6 months prior to surgery predicative of increased comorbidity and length of stay following surgical resection of pancreatic tumors. Inclusion of these data if available and any associations with the complication data reported by the authors in this patient cohort would improve the manuscript.

5. Were CRP measurements taken in patients with pancreatic cancer prior to surgery? If so, please provide these data in Table 1.

6. While there is a Table 2 and Table 4, there is not Table 3. Please correct table numbering.

7. Please address why the percent distribution of patients does not add up to 100% for some variables (eg., Cancer Stage, lines 73-74).

Comments on the Quality of English Language

Overall minor editing of the manuscript for spelling and to make sentences more concise and coherent would improve the readability. For instance, "changes in amino acid metabolism can affect cancer cell," (lines 39-40), clarification is needed to state what is being affected in cancer cells by changes in amino acid metabolism. "Volonteers" is misspelled in Table 1.

Author Response

We thank you for your thoughtful comments and suggestions for improving our manuscript.

Replying to comments:

1. Understanding the patient variables in the methods (Lines 72-79) takes a lot of time when listed in paragraph form. Consider putting the data contained in lines 72-79 in table format. This would make it faster for a reader to interpret. 

Thanks for corrections. We have added the table 1 in the patient description section (Line 114-117).

2. While the authors mention the small sample size of patients in this study, metabolite measurements (Table 4) and complication severity data (Figure 2) separated into male and female groups would enhance the manuscript. For instance, overall elevated PhLA levels in patients are associated with severe complications (Figure 2) and increased ICU stay (Table 4), it would be interesting to know the sex distribution of these groups too. 

We cannot increase the sample size for this article, but we plan to continue our research. We stratify the patients by relevant clinical and demographic characteristics, such as gender, age, stage of cancer, tumor location.  It’s did not affect our results, except for the location of the tumor, which is reflected in table 4, in the text on lines 234-239 and in appendix tables 1 and 2.

  1. In addition to Comment 2, it would also be interesting to know if patient physical status before surgery (ASA categorization) is associated with complications after surgery and/or ICU stay. This would enhance the study since complicating factors such as obesity could have an effect on these data.

Given your recommendations to consider a number of clinically important confounders when comparing patient data, our research team performed a number of additional statistical tests to analyze independent groups of patients and presented them in Table 4  and Table 1. A. (Appendix Table 1. A.).

  1. Do the authors have any additional patient data including preoperative body composition measurements (eg., CT scans) or 1, 3, or 6-month weight loss prior to surgery? Previous studies have correlated >5% body weight loss up to 6 months prior to surgery predicative of increased comorbidity and length of stay following surgical resection of pancreatic tumors. Inclusion of these data if available and any associations with the complication data reported by the authors in this patient cohort would improve the manuscript.

Unfortunately, this is a pilot study and one of the limitations is that we only have information about patients when they are admitted to the clinic immediately before surgery. We took into account such parameters as gender, age, stage of cancer and saw that this did not affect our results, which is reflected in table 4 and in appendix tables 1 and 2.

  1. Were CRP measurements taken in patients with pancreatic cancer prior to surgery? If so, please provide these data in Table 1.

We have added these data and the CRP level is also presented in Appendix Table 1.

  1. While there is a Table 2 and Table 4, there is not Table 3. Please correct table numbering.

Thanks for corrections. We have added the table 1 in the patient description section and corrected the numbering of all tables in the manuscript.

  1. Please address why the percent distribution of patients does not add up to 100% for some variables (eg., Cancer Stage, lines 73-74).

Thanks for corrections. We have corrected the patient descriptions and transferred this information to Table 1.

Comments on the Quality of English Language

Overall minor editing of the manuscript for spelling and to make sentences more concise and coherent would improve the readability. For instance, "changes in amino acid metabolism can affect cancer cell," (lines 39-40), clarification is needed to state what is being affected in cancer cells by changes in amino acid metabolism. "Volonteers" is misspelled in Table 1.

Thank you for correcting typos in the text, we corrected them and checked the spelling again